

# Simulating physiological flexibility in the acute glucocorticoid response to stressors reveals limitations of current empirical approaches

Conor Taff[1,2]

[1] Department of Ecology and Evolutionary Biology, Cornell University, Ithaca, NY, United States
[2] Lab of Ornithology, Cornell University, Ithaca, NY, United States

## ABSTRACT

Wild animals often experience unpredictable challenges that demand rapid and flexible responses. The glucocorticoid mediated stress response is one of the major systems that allows vertebrates to rapidly adjust their physiology and behavior. Given its role in responding to challenges, evolutionary physiologists have focused on the consequences of between-individual and, more recently, within-individual variation in the acute glucocorticoid response. However, empirical studies of physiological flexibility are severely limited by the logistical challenges of measuring the same animal multiple times. Data simulation is a powerful approach when empirical data are limited, but has not been adopted to date in studies of physiological flexibility. In this article, I develop a simulation that can generate realistic acute glucocorticoid response data with user specified characteristics. Simulated animals can be sampled continuously through an acute response and across as many separate responses as desired, while varying key parameters. Using the simulation, I develop several scenarios that address key questions in physiological flexibility. These scenarios demonstrate the conditions under which a single glucocorticoid trait can be accurately assessed with typical experimental designs, the consequences of covariation between different components of the acute stress response, and the way that context specific differences in variability of acute responses can influence the power to detect relationships between the strength of the acute stress response and fitness. I also describe how to use the simulation tools to aid in the design and evaluation of empirical studies of physiological flexibility.

## INTRODUCTION

Animals live in a dynamic environment in which they regularly encounter unpredictable challenges. Successfully navigating these challenges often requires the ability to rapidly adjust behavior and physiology to match current conditions. For vertebrates, the glucocorticoid mediated stress response plays a major role in coordinating these changes when stressors are encountered (*Sapolsky, Romero & Munck, 2000*; *Wingfield et al., 1998*)

Corresponding author
Conor Taff, cct63@cornell.edu

and similar rapid response systems mediate changes in other taxa (*Taborsky et al., 2021*). Because of the central role that this response plays in coping with challenges, a great deal of research effort over the past 15 years has focused on understanding whether between-individual differences in the magnitude of this response predict coping ability and, ultimately, fitness (*Breuner, Patterson & Hahn, 2008*; *Schoenle et al., 2020*).

More recently, a series of conceptual articles have asked whether the degree of within-individual variation in glucocorticoid modulation (*i.e.*, endocrine flexibility) across different contexts or in response to different stressors might also be an important predictor of performance (*Hau et al., 2016*; *Lema & Kitano, 2013*; *Taff & Vitousek, 2016*; *Wada & Sewall, 2014*). Perhaps the major limit to empirical progress, especially for within-individual variation, is the logistical difficulty of accurately characterizing the functional shape of the acute physiological stress response for an individual during a single acute response and across multiple acute responses occurring under different conditions. Often these measures are strictly limited by the number of samples that can safely be taken from an animal during a single capture and the number of repeated captures that are possible (but see *Koolhaas et al., 2011*). Given these limitations, data simulation is a powerful tool that could complement empirical work in this area, but that has not yet been applied to studies of endocrine flexibility.

Several recent articles have suggested that physiologists interested in endocrine flexibility should adopt a within-individual reaction norm approach (*e.g.*, *Hau et al., 2016*; *Taff & Vitousek, 2016*). This approach has been widely adopted in studies of behavioral flexibility where statistical methods and empirical progress have developed synergistically (*e.g.*, *Araya-Ajoy, Mathot & Dingemanse, 2015*; *Dingemanse et al., 2010*; *Westneat, Wright & Dingemanse, 2015*). This field has also benefited from simulation studies to evaluate optimal study design (*van de Pol, 2012*) and packages that can create artificial datasets with desired patterns of between, within, and residual variance to evaluate the consequences of different patterns of variation on the ability to detect effects (see the SQuID package, *Allegue et al., 2017*). While these approaches are powerful, they have proven difficult to apply directly to endocrine flexibility data for two reasons. First, simulation studies suggest that successfully modeling within-individual variation in flexible traits using an hierarchical modeling framework often requires a level of repeated sampling that is possible for many behaviors (especially when collected autonomously), but that is currently not possible for most studies of endocrine flexibility, because it would require sampling of many separate glucocorticoid responses per individual. Second, many behavioral articles focus on somewhat discrete measures (*e.g.*, aggression score or activity level), whereas for acute glucocorticoid responses, the functional shape of the response itself may be the important trait. Fully describing the functional shape of a single acute glucocorticoid increase may require many samples in close succession, but for small vertebrates logistical and ethical constraints mean that it is rarely possible to take more than a few samples during the course of a single acute response.

The function valued trait (FVT) framework is an alternative approach that explicitly considers the functional shape of a biological response (*Gomulkiewicz et al., 2018*; *Kingsolver, Diamond & Gomulkiewicz, 2015*; *Stinchcombe, Kirkpatrick & Function-valued*
*Traits Working Group, 2012*). While FVT approaches have been suggested for studies of endocrine flexibility, I am not aware of any articles that have applied this framework to empirical data on acute glucocorticoid responses, probably because sufficient data are not available. Conceptually, however, this approach is a better match to the acute glucocorticoid response, because the shape of a response curve is explicitly considered as the phenotypic trait of interest. In some cases, it may make sense to estimate particular parameters of the curve (*e.g.*, maximum rate of increase and maximum value reached) and then treat those parameters as phenotypic values for downstream analysis, although statistical methods also exist to analyze the shape of the entire curve directly without the need to extract discrete parameters (*Kingsolver, Diamond & Gomulkiewicz, 2015*). This approach has been used to study a variety of phenotypes where values can be measured continuously or pooled across many individuals from the same group to accurately estimate the shape of a curve (see Table 1 in *Stinchcombe, Kirkpatrick & Function-valued Traits Working Group, 2012*). Applying the technique to endocrine flexibility at the within-individual level faces the same empirical challenges described for within-individual reaction norms above, such as the need for repeated sampling of individuals and high temporal resolution of samples within individual physiological responses. Note that FVT and within-individual reaction norms approaches are not necessarily incompatible, but they have largely developed separately.

The recognition that characterizing the functional shape of an acute stress response is challenging goes back to the earliest studies conducted in wild animals. Early studies often employed various control groups and sampled individual animals at a variety of time points over a long period in order to describe the full response curve for a particular group (*e.g.*, a species or a breeding stage, *Wingfield, Vleck & Moore, 1992*). These validations were considered essential to characterize key parameters of the acute response for each group being studied (*i.e.*, baseline, rate of increase, maximum level, time of peak, and area under the curve; J. Wingfield, 2021, personal communication). Indeed, there is a long and rich history of empirical work characterizing differences in each aspect of the acute glucocorticoid response (*Breuner, Wingfield & Romero, 1999*; *e.g.*, *Cockrem & Silverin, 2002*; *Love, Bird & Shutt, 2003*; *Wingfield, Vleck & Moore, 1992*) and empirical data has contributed to a variety of conceptual models of acute glucocorticoid regulation (*Romero, Dickens & Cyr, 2009*; *e.g.*, *Wingfield et al., 1998*). More recently, mathematical models have been used to generate predictions about how flexibility and variation in different parts of the response system might respond to particular conditions (*Grindstaff et al., 2022*; *e.g.*, *Luttbeg et al., 2021*; *Taborsky et al., 2021*). However, the challenge of estimating these parameters and of decomposing within- and between-individual variance becomes much more difficult when trying to fully describe the response for an individual animal rather than for a group, because glucocorticoids can often only be measured at two or three time points and only a small number of times per animal (*e.g.*, *Vitousek et al., 2018*). Because these studies require an estimate for each individual, the solutions used by older studies that added additional animals to allow for sampling at more time points are not available.

For individual based studies, the most common approach to this problem is to standardize measurements as much as possible by measuring animals at the same time of

the day during the same context, and by taking blood samples at standard times (often <3 and 30 min after capture) to characterize baseline and stress-induced glucocorticoids. This standardization allows for comparison between individuals, but in some cases it may also completely obscure the ability to detect variation in certain characteristics of the acute response curve. For example, if the speed (rate of initial increase or time required to reach maximum) and scope (maximum value) of the acute response vary independently, samples taken at only two time points cannot accurately capture variation in either parameter (*Taff, Wingfield & Vitousek, 2022*). Indeed, several discussions in recent years about methods such as the '3 minute rule' and the relative merits of 'area under the curve' *vs* time point measures of glucocorticoids are fundamentally related to a recognition of the importance of understanding variation in the functional shape of stress responses and whether different components of that shape covary within individuals (*e.g.*, *Cockrem & Silverin, 2002*; *Small et al., 2017*). While a great deal of empirical work has focused on characterizing the rate of initial increase (*Cockrem, 2005*; *Cockrem, Potter & Candy, 2006*), there has been relatively little work on understanding individual differences in the time required to reach maximal levels, and this attribute of speed is harder to estimate (*Taff, Wingfield & Vitousek, 2022*).

One of the characteristics of both the within-individual reaction norm and FVT literature is that empirical work has proceeded in very close coordination with simulation and statistical method development. In contrast, studies of endocrine flexibility often point to these methods, but do not address the ways that the particular logistical challenges of hormone measurement, such as the difficulty of repeatedly collecting samples within- or between-individuals, might necessitate different empirical approaches. I believe this is one reason that there are currently more conceptual articles arguing for a reaction norm approach to endocrine variation than there are empirical articles actually applying the approach (but see, *Fürtbauer et al., 2015*; *Houslay et al., 2022*; *Lendvai et al., 2014*; *Taff, Wingfield & Vitousek, 2022*). While many of the tools developed in these related fields are transferable, studies of physiological flexibility would benefit from a focus on analysis development and testing that explicitly incorporates the particular details and challenges of these questions. One way to accomplish these goals is to integrate empirical work with simulations, but to my knowledge no studies of physiological flexibility have developed simulations of the acute stress response that address the issues discussed above.

Data simulation is a powerful approach for several reasons. Because true parameter values (*e.g.*, maximum glucocorticoid level) are known, it is possible to evaluate how well different study designs and analytical choices perform in recovering true patterns and how sensitive those designs are to different assumptions. Thus, simulation can tell us whether the study designs we use can in principle detect the patterns we predict given realistic effect sizes. Simulated data can also identify conditions under which current study designs will perform well or poorly. For example, if simulations suggest that the baseline paired with stress-induced paradigm only works well when the speed and scope of responses are positively correlated, then empirical work could seek to determine the degree of correlation for a particular study system as justification for the approach. This ability to highlight key assumptions and create data sets with known properties has the potential

to both provide insight into physiological flexibility directly and to guide empirical work by improving study design and identifying key areas for subsequent sampling. In the rest of this article, I develop a simple simulation of acute physiological stress responses and then briefly illustrate several possible applications of the simulation.

The goal of this simulation is to provide a flexible tool that can produce realistic datasets of physiological flexibility for a variety of different systems and scenarios. As such, there are many possible applications and here I briefly highlight a few possibilities. Initially, I demonstrate that the simulation can produce datasets that are qualitatively similar to empirical data on acute glucocorticoid responses. Next, I explore three specific scenarios with the simulation that highlight challenges to empirical progress in this field. First, I ask how well maximum glucocorticoid measures can be estimated with a single time point measure in populations that differ systematically in features of the response. Second, I explore how different patterns of covariation between maximum glucocorticoids and one measure of speed (the time to reach maximum) influence the ability to reliably measure either component of the response. Third, I ask how different amounts of between- and within-individual variation in maximum glucocorticoid levels impact the ability to detect known relationships between glucocorticoids and fitness. These questions are all important for understanding physiological flexibility and are difficult to fully address empirically, but this is by no means an exhaustive list of the questions or possible parameter permutations that could be explored with simulation. Finally, I demonstrate how the simulation tools can be used to evaluate the performance of different experimental designs given a set of logistical limitations (*e.g.*, number of samples that can be collected) and discuss how this procedure could be used as a planning tool to increase the power and reproducibility of future empirical work.

## MATERIALS AND METHODS

### Description of the simulation

I developed a set of functions in R version 4.0.2 (*R Core Team, 2020*) to generate acute physiological response curves. This simulation makes no assumptions about the mechanistic process that results in the shape of a glucocorticoid response. Rather, parameters are sampled to generate curves that are similar in shape and degree of variation to empirically observed responses (Fig. 1). This simulation is designed to create data sets with realistic structure that can be used to better design and plan studies of physiological flexibility, to evaluate power of current study designs, and to evaluate the sensitivity of sampling regimes to any number of modifications to the shape of glucocorticoid response curves (*e.g.*, changing covariation patterns between different features of the response). I explore a small number of scenarios in the next section, but I expect that many other scenarios can be addressed with these tools. For illustration purposes, I refer to simulated glucocorticoid responses, but the simulation applies equally well to any physiological mediator of a rapid response. The code for the complete simulation along with full documentation of each function and argument can be accessed in a GitHub repository with a current version permanently archived on Zenodo
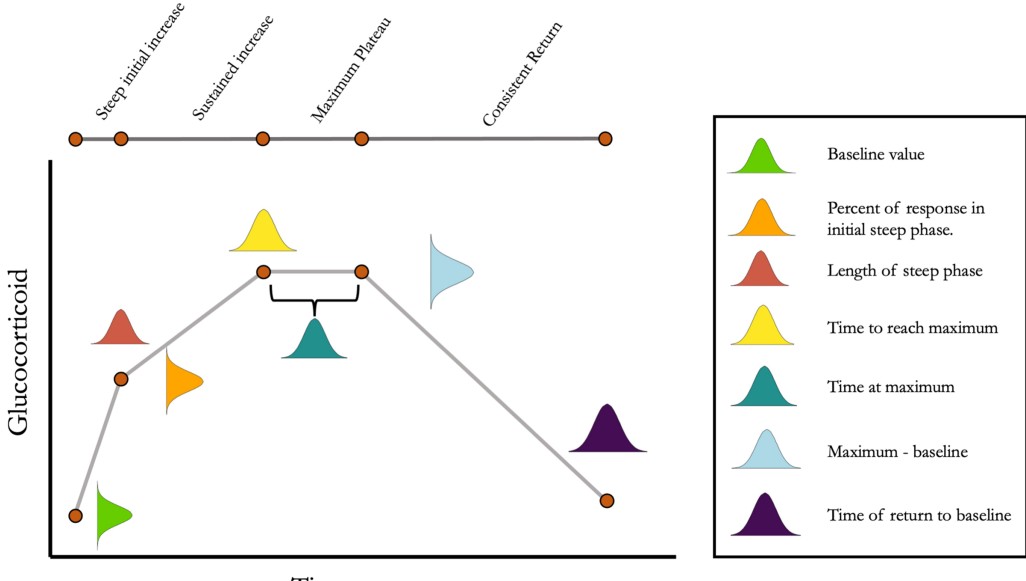

**Figure 1 Conceptual illustration of the structure of the simulation.** For each simulated animal, seven parameters are sampled from a multivariate normal distribution. Together, these seven parameters define the turning points in an acute response curve. The mean and standard deviation for each parameter can be set along with the degree of covariation between each pair of parameters. Note that the simulation can easily be simplified as desired by setting some parameter mean or standard deviations to zero.

(https://doi.org/10.5281/zenodo.6784207). The package can be installed directly within the R environment using the following command.

devtools::install_github("cct663/simcoRt")

The simulation is constructed as two main functions with several minor functions for downstream analysis. Detailed descriptions of the arguments to each function are included with the package documentation. Briefly, function cort_sim1 samples the parameters shown in Fig. 1 from an arbitrary number of animals. These parameters are sampled from a multivariate normal distribution with user specified mean, variance, and covariance for each parameter. By default, maximum glucocorticoid values are sampled from a normal distribution on the log scale and then exponentiated to determine absolute values. This results in a right skewed maximal glucocorticoid distribution on the absolute scale that is typical of many empirical datasets, but users can easily specify any parameters to be sampled from a normal or log normal distribution as required to match the characteristics of a particular study system. For the purposes of the simulation, I consider the sampled parameter values to be the 'true,' unobserved, phenotype of the animal (setting aside the question of whether or not a 'true' physiological phenotype exists).

A second function, cort_sim2, starts with a population of animals generated from cort_sim1 and samples observed acute glucocorticoid responses an arbitrary number of times for each animal. Two sources of variation in the observed relative to true parameter values can be specified. First, within-individual variation in expression is represented by specifying what amount of variation in the observation of each parameter is determined

by the true value and what amount is determined by an additional randomly sampled response, based on the population parameters (this additional sampling maintains the user specified covariance structure of the population). After sampling the parameters, values are interpolated for each 1 min time point and a localized regression is fit to create a smoothed curve that represents the observed glucocorticoid response. From this expressed response, individual data points are then collected at user specified times that would reflect an empirical study design (*e.g.*, 1, 30, and 60 min). Additional noise can be added to these data points to represent measurement error (*e.g.*, assay error). This simulated dataset can then be treated as the input for any desired analyses and statistical approaches, while maintaining the ability to compare results to the 'true' values used in the simulation.

The function also generates a simple simulated performance (*e.g.*, fitness) measure, based on the underlying true parameter values sampled for each individual in the population (*e.g.*, their baseline and maximum glucocorticoid value). The single fitness measure per animal is determined by allowing the user to specify the relative degree to which unmeasured traits plus each true parameter contribute to fitness outcomes. Data reflecting the true phenotypic values, the repeated expression of acute responses, and the observed time points can then be used in downstream analyses with any standard statistical approaches or software. For example, a user could perform an analysis to ask whether a known relationship between fitness and a particular true parameter is recovered in a study that includes only measures taken at particular time points. An additional convenience function summarizes the output of a simulation run in a multi-panel plot (Fig. 2).

Finally, given recent interest in estimating the repeatability of glucocorticoid regulation (*Cockrem, 2013*; *Hau et al., 2016*; *Taff, Schoenle & Vitousek, 2018*), I also included a function that takes input from cort_sim2 and calculates the observed repeatability of several measures using package rptR (*Stoffel, Nakagawa & Schielzeth, 2017*). Full details are included in the package documentation, but this function returns repeatability for each individual time point specified in the down sampled data set, profile repeatability (*Reed, Harris & Romero, 2019*), and repeatability for area under the curve (AUC) calculated as both increase ($AUC_I$) and ground ($AUC_G$) approaches (*Pruessner et al., 2003*). For each AUC measure, the function returns repeatability for the full time course, for an estimate using only the observed values in the down sampled data set, and for the full data set constrained to the time period encompassing the observed data points. Simple plots illustrating repeated samples from the same individuals are also returned by default. I do not develop an example of repeatability in this manuscript, but the functions here could be used to determine the impact of different study design choices on repeatability estimates.

## Methods for each simulated scenario

Each of the scenarios described here uses the basic simulation functions described above with input parameters adjusted to address the question of interest. A complete set of reproducible code to create all of the examples presented here is available on GitHub and permanenty archived at Zenodo (https://doi.org/10.5281/zenodo.6784203).
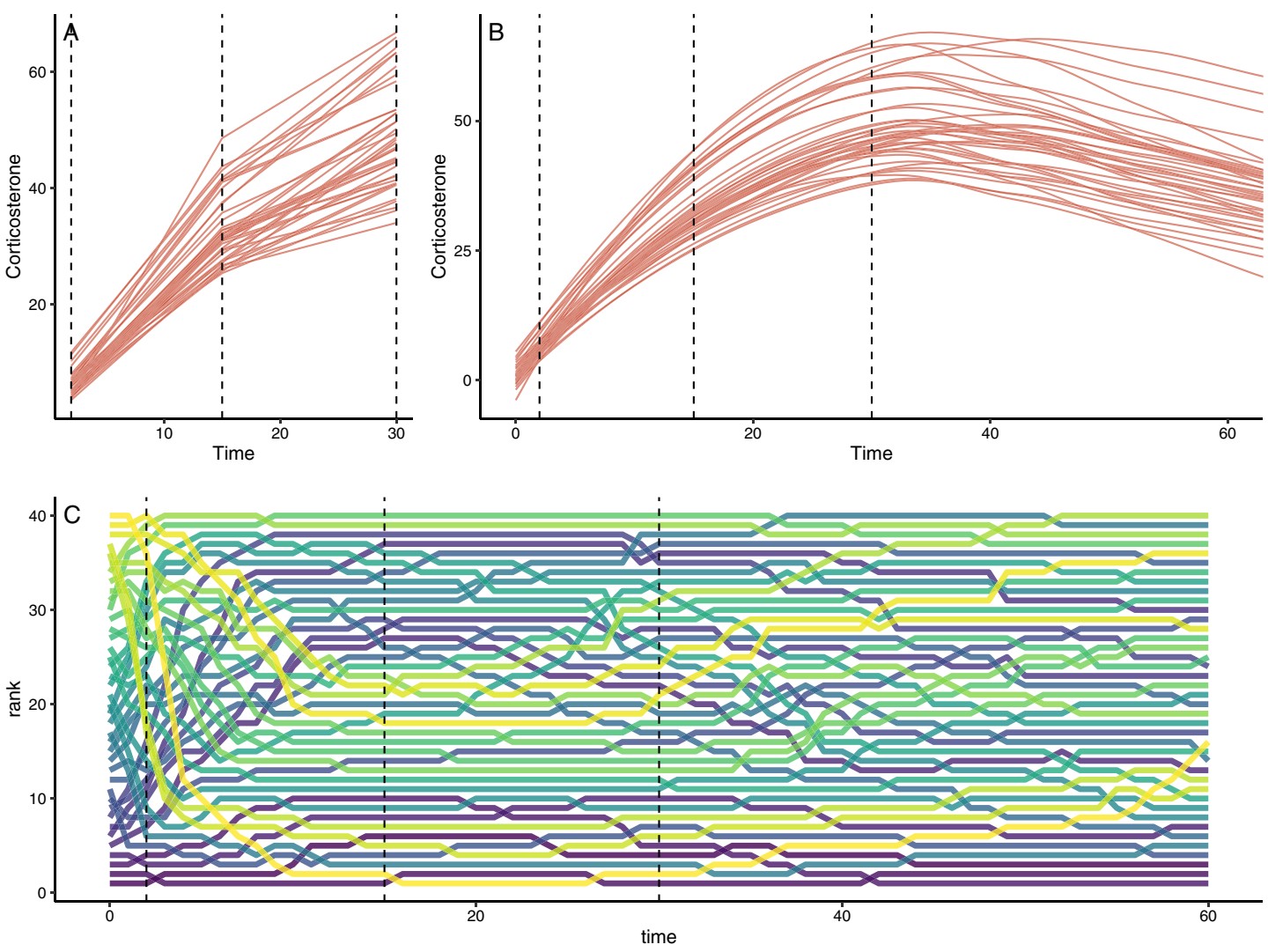

**Figure 2 Example of simulation output with default settings.** (A) Shows the downsampled data set for this run with samples collected at 1, 15, and 30 min in this case. (B) Shows the full observed response curve for each animal. (C) Shows the rank order of glucocorticoid level at each time point for each animal. In each panel, the vertical dashed lines represent the three time points that might have been measured in a typical empirical study. Note that individuals in the top panels do not match perfectly because measurement error is added to the downsampled dataset in (A).

## Scenario 1: simulating empirically parameterized data

In order for simulation to be useful, we should be able to create artificial datasets that have similar characteristics to empirical data for different systems. Simulating realistic data provides a starting point for evaluating different study designs and the consequences of changes in different assumptions or parameters. Simulating realistic data is also useful because it can aid in study design or be used as a basis for pre-registered reports that demonstrate the feasibility of a planned study before data are ever collected. Simulated data can be created and entered in a complete analysis pipeline, with empirical data substituted later. In addition to helping to design better studies, this approach has the advantage of

increasing the transparency and reliability for studies of physiological flexibility, by making analysis choices and predictions clear before data are collected.

To demonstrate this utility, I attempted to create simulated datasets that recreate the characteristics of the empirical data presented in *Koolhaas et al. (2010)*. As part of that study, a series of corticosterone measurements were collected during and after an acute stressor from 14 laboratory rats *Rattus norvegicus* using permanently implanted jugular vein canulae. I extracted data from Figure 6 in *Koolhaas et al. (2010)* using WebPlotDigitizer (https://automeris.io/WebPlotDigitizer/) and then simulated data using the functions described above starting with the input values calculated directly from the empirical data. I then compared patterns of variation and population level response curves for the empirical and simulated data.

## Scenario 2: accurately measuring a single glucocorticoid trait

Single time point measures of glucocorticoids are often interpreted as representing meaningful variation between individuals. For example, variation in the level of glucocorticoids after 30 min of standardized restraint is typically interpreted as variation in the magnitude of the stress response (*Taff, Zimmer & Vitousek, 2019*). However, this interpretation rests on assumptions that are rarely explicitly tested with empirical data. For example, the time chosen to take a stress-induced sample is often assumed to be either at the species peak or during a plateau period after the species peak. In some early studies, great care was taken to determine an average population level peak time (*e.g.*, *Wingfield, Vleck & Moore, 1992*), but many studies adopt the widely used 'standard' time of 30 min post capture without extensive validation (compiled in *Vitousek et al., 2019*).

While there is a general assumption that sampling later than the peak is acceptable (and perhaps preferable) because animals will be sampled during a relatively stable high plateau, there is little empirical data to evaluate this assertion or to determine how much under or overshooting the species peak timing might influence inferences. Furthermore, even when the average peak timing is well established, differences in the amount of between-individual variation in the time to reach the peak or in peak values are common across species and even in different life history stages within species (*Wingfield, Vleck & Moore, 1992*). Some studies focus instead on the rate of initial increase in glucocorticoids with a sample taken at 10 or 15 min after disturbance (*e.g.*, *Cockrem, 2005*; *Cockrem, Potter & Candy, 2006*; *Love, Bird & Shutt, 2003*), often well before peak levels are reached. The combination of different patterns of within- and between-individual variation with the exact time(s) chosen for sampling could have consequences for the accuracy of point estimates taken at any single time point, but these questions cannot be addressed directly with empirical datasets where the true underlying values of each individual are unknown.

I simulated a simple scenario to explore the consequences of variation in each of these parameters on the accuracy of estimating between individual differences in maximally expressed glucocorticoids during an acute response. For purposes of this illustration, I consider a single study design in which animals are sampled at 30 min. Using this design as a starting point, I systematically vary (i) the timing of the population average peak

(15, 30, or 45 min), (ii) the amount of variation in maximum glucocorticoid levels reached (standard deviation (SD) of 1 to 12 ng/$\mu$l), (iii) and the amount of variation in the number of minutes taken to reach peak levels (SD of 1 to 20 min). All other variables in the simulation are constrained to be invariant between individuals in the population (*e.g.*, all individuals have identical baseline glucocorticoids in this case), though I consider cases in which multiple aspects of the rapid response are correlated with each other in the next scenario. I included moderate within-individual variability and a small amount of assay error across all iterations. For each combination of parameters, I simulated 200 animals and estimated the $R^2$ value from a regression of the observed estimates of glucocorticoid levels at 30 min to the true known values. This simulation is likely a best case scenario for detection because it eliminates many sources of variation or noise that would be present in real data, but it illustrates the effect of variation in these three key parameters even when the exact same sampling design is employed.

## Scenario 3: exploring covariance between response components

In reality, fully characterizing the acute glucocorticoid response requires more than identifying just the maximum value reached. Individuals may differ in baseline levels, rate of initial increase, the speed of reaching the maximum level, time spent at maximum, and the speed of return to baseline. Moreover, each of these components of the endocrine response could be positively or negatively correlated with each other within and between individuals. In these cases, measurements taken at particular time points contain information about multiple aspects of the response and without additional information it may be difficult to know what trait is being measured. The fact that each of these traits might be important and that they might covary has been discussed in a general sense (*e.g.*, Baugh et al., 2013), but simulations are uniquely powerful for exploring under exactly what conditions time point measure of glucocorticoids can or cannot be used as indicators of these traits.

To illustrate this point, I explored the consequences of variation in the correlation between and relative amount of variation in just two aspects of the acute stress response: the maximum glucocorticoid level reached and the time required to reach the maximum level. For simplicity, I refer to the 'speed' of the response, but note that other aspects, such as the rate of initial increase, could also be considered as variation in the speed of response. When considering these two traits, a population of animals could plausibly display one of three patterns. Individuals that reach their maximum value faster might also reach higher values (simulation correlation = −0.6). Alternatively, the speed and maximum values might vary independently (correlation = 0). Finally, individuals that are faster responders might max out at lower glucocorticoid values (correlation = 0.6). While many researchers in this field might have intuitions about which of these scenarios is most likely to prevail, there is very little empirical data available to actually determine which is most common. Moreover, regardless of the specifics for this particular correlation, the general pattern and considerations presented here will apply in similar ways to correlations between other aspects of the acute stress response.

Using these three simulated populations as a starting point, I asked how well glucocorticoid values measured at one timepoint reflected true trait values. For each population I set an average population level speed of 30 min with other values in the simulation set at their default value. For every time point from 0 to 35 min I fit two simple linear regressions of the measured value on the true speed and maximum value and extracted the $R^2$ value from the model. I repeated this simulation for all populations 50 times with 100 individuals sampled from the population each time. Finally, I repeated the entire set of simulations with each combination of low and high between-individual variation in the speed or maximum values (variation in speed: low = 2 minute SD, high = 12 minute SD; variation in maximum: low = 1 ng/$\mu$l SD, high = 10 ng/$\mu$l SD).

## Scenario 4: detecting links between fitness and responses

A common goal of recent studies is to establish whether variation in glucocorticoids is associated with fitness or some proxy for fitness (*Schoenle et al., 2020*). While there has been a great deal of discussion about the extent to which these relationships might differ with life history characteristics or between breeding stages, there has been relatively little consideration of the way that methodological limitations might limit the ability to detect these relationships even when they exist.

Here, I imagine a simple scenario in which the 'true' maximum glucocorticoid level during an acute response explains 80% of the variation in fitness (clearly this is unrealistically high, but it is chosen for illustration only). I next construct a study in which researchers measure 50 individuals using a typical stress-induced (30 min) sampling protocol. For simplicity, I set the other parameters in the simulation at their default values. Keeping the study design constant, I ask whether the glucocorticoid-fitness relationship can be recovered for two hypothetical populations that have low or high between-individual variation in maximum glucocorticoid levels. Using these populations, I asked how the ability to detect glucocorticoid-fitness relationships changed with different amounts of within-individual variation in acute response expression. For each combination of parameters, I simulated 50 populations and fit a simple linear regression model with observed glucocorticoid levels at 30 min as a predictor of fitness to ask whether the true glucocorticoid-fitness relationship was recovered.

## Scenario 5: designing optimal sampling strategies

One of the major benefits of simulating glucocorticoid response curves will be the ability to design optimal sampling strategies before data are collected. A simulation can be constrained to match any real world limitations (*e.g.*, maximum number of samples possible per individual) and then explored to determine how to best allocate sampling resources. The specifics of this task will vary considerably with the study system and question being addressed, but here I illustrate one possible application. Consider an experiment in which the acute glucocorticoid response of a treatment group and control group are compared after some experimental manipulation. The details of the manipulation are unimportant here, but suppose that the prediction is that this manipulation should result in a difference in the speed of the corticosterone response

between our two groups, such that the treatment group will reach it's maximum glucocorticoid value faster than the control group, but will not differ in the maximum value itself. I have implemented this difference by simulating two populations in which the treatment group has a steeper initial slope and also reaches the maximum value faster. Any number of possible hypotheses for a particular study system could be specified following a similar approach.

Using these simulated groups, I asked how well different study designs could detect the differences. Here we can impose any logistical constraints relevant to the study system. As an example, in this case we can only sample a maximum of 20 individuals per group, we can only sample each individual once post-treatment, and during that single sampling event we can take a blood sample at a maximum of three different time points, resulting in a total of 120 data points. Given these constraints, I compare several sampling designs. First, I explored 'standard' sampling strategies that are typical of empirical studies in this field that sample each individual animal at either (i) 1, 30, and 60 min, (ii) 1, 15, and 30 min, or (iii) 1, 15, and 60 min. Second, I explored two alternative sampling approaches that are not typically used in empirical work but might better capture the entire functional shape of response. These included, (iv) a study in which three sampling times between 1 and 60 min are randomly chosen for every animal, and (v) a study in which three sampling times are randomly chosen for each animal, but weighted more heavily around the range of times when maximum levels are expected to be reached for the population.

For illustration purposes I sampled directly from the 'true' response curves in this example so that there is no additional measurement error added. To evaluate these schemes I compare estimates of the acute response curve for each group to the 'true' known curves. Note that a more complete analysis of a sampling schemes performance should include many more iterations and full statistical comparisons, but the details here will be highly dependent on the study system and goals, so I provide this simple example to illustrate the approach rather than to make any more widely applicable conclusions.

## RESULTS

### Scenario 1: simulating empirically parameterized data

The simulation functions were able to produce a new synthetic dataset that has similar variation and patterns to the empirical data in Koolhaas et al. (2010) (Fig. 3A). The average population wide response curve shape also closely matched the empirical data (Fig. 3B). In this case, the plotted simulation data include the same number of animals sampled at the same time points as the empirical data, but these sampling points and total sample size can easily be changed as desired. The parameterized simulation can now be used to test the sensitivity of any number of experimental designs before additional data is collected, such as different sample sizes or sampling time points.

### Scenario 2: accurately measuring a single glucocorticoid trait

The amount of between-individual variation in maximum glucocorticoid values has a profound effect on the ability to detect true maximal levels with samples taken at 30 min (Fig. 4). A single sample taken at 30 min was highly correlated with true maximal
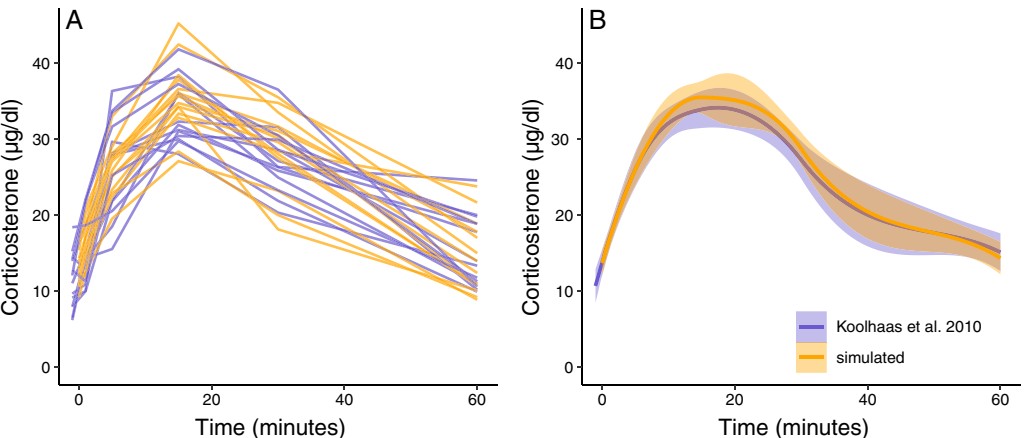

**Figure 3 Simulated dataset matching empirical data.** (A) Shows the acute corticosterone response for measured (blue) or simulated (orange) rats measured at five time points. (B) Shows the mean and standard error of the two datasets. Empirical data are extracted from *Koolhaas et al. (2010)* Fig. 6 using the WebPlotDigitizer tool.

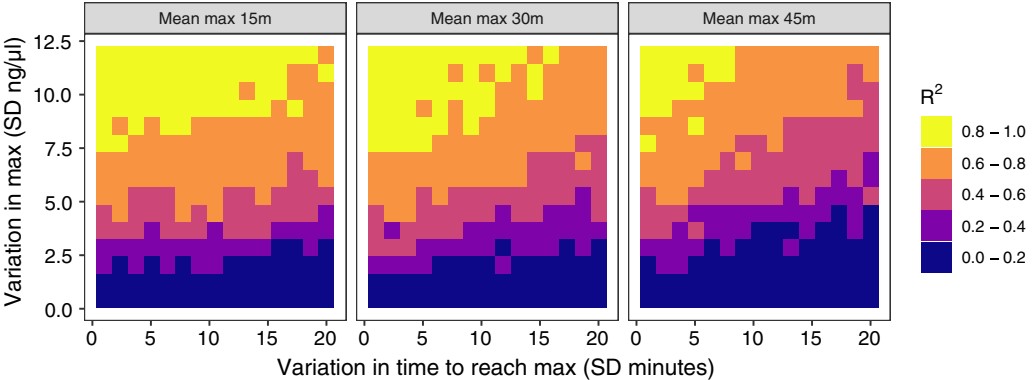

**Figure 4 Results of simulation runs with different amounts of between-individual variation in the time to reach maximum glucocorticoid levels and in the maximum level reached.** Simulations are run with samples taken at 30 min on populations with an average peak time of 15 min (left), 30 min (center), or 45 min (right). Each grid cell is the $R^2$ value from the regression of observed glucocorticoids at 30 min to true maximum levels in a simulation of 200 individuals.

glucocorticoid levels when between-individual variation was high, but with low between-individual variation in maximum a single sample was uninformative. I simulated a wide range of variation that may include unrealistically high and low values, but the result highlights the importance of considering the amount of variation expected in a study population. There is a weaker, but still substantial impact of variation in the time taken to reach maximum values on the accuracy of estimates in this simulation. Greater variation in the speed (time to reach maximum) of the response reduces the accuracy of estimates of maximal values. Finally, the timing of sampling relative to the average population peak timing also influences accuracy. Measuring after the average peak time results in the most accurate estimates across a range of parameter values, while measuring
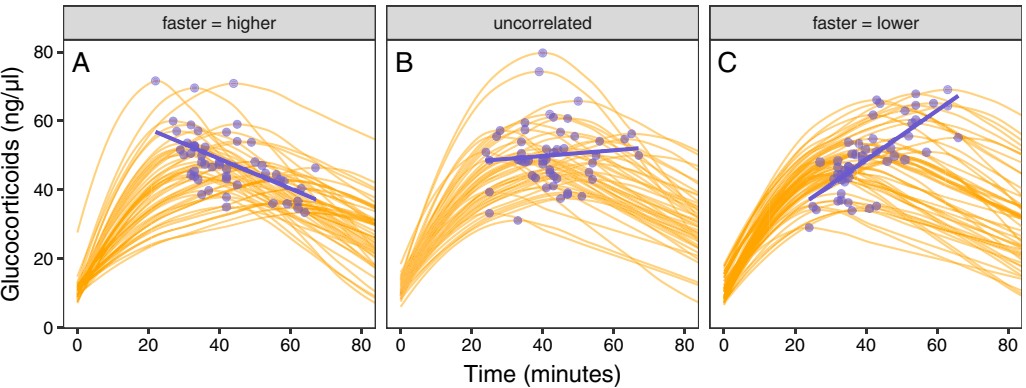

**Figure 5 Varying correlation between speed and scope of response.** Simulated glucocorticoid responses in which the maximum value and response speed are positively correlated (A), uncorrelated (B), or negatively correlated (C). Orange curves show the full response for each individual. Blue points show the maximum value and time to reach maximum for each individual. Blue lines are simple linear regressions of speed and maximum value for each group. For clarity, only the first 40 individuals in each simulated dataset are plotted.

before the average peak time produces the least accurate measures, particularly when there is also high variation in the time to reach maximum values between individuals.

## Scenario 3: exploring covariance between response components

Simulated populations had either a positive correlation between the time required to reach maximum values and the maximum glucocorticoid level reached (Fig. 5A), no correlation (Fig. 5B), or a negative correlation (Fig. 5C). Using these populations, the time that samples were taken at, relative amount of variation in speed and maximum, and degree of correlation between the speed and maximum all had substantial impacts on the ability to infer true trait values from single time point glucocorticoid measures (Fig. 6). Neither speed nor maximum traits could be assessed accurately when between-individual variation in both traits was low (Fig. 6A). Overall, accurately assessing variation in speed was much harder—if not impossible—with single measures.

It was only possible to accurately estimate speed when high between-individual variation in speed was coupled with low variation in maximal values, but this situation may be rare in natural populations. When speed was tightly correlated with maximum (Fig. 6D) it was sometimes possible to attain reasonable estimates of speed (Figs. 6C and 6D), but when speed was not correlated with maximum, single measures were not good indicators of variation in speed (Figs. 6A, 6C and 6D). Finally, measuring variation in maximum values was much easier under many conditions (Figs. 6C and 6D), but the accuracy of assessment of maximum values was also negatively impacted by variation in speed and the degree of this impact differed depending on the correlation between the two traits (Fig. 6).

## Scenario 4: detecting links between fitness and responses

Several patterns can be identified by examining the results of this simulation. First, as specified by the simulation parameters, the correlation between the true maximum

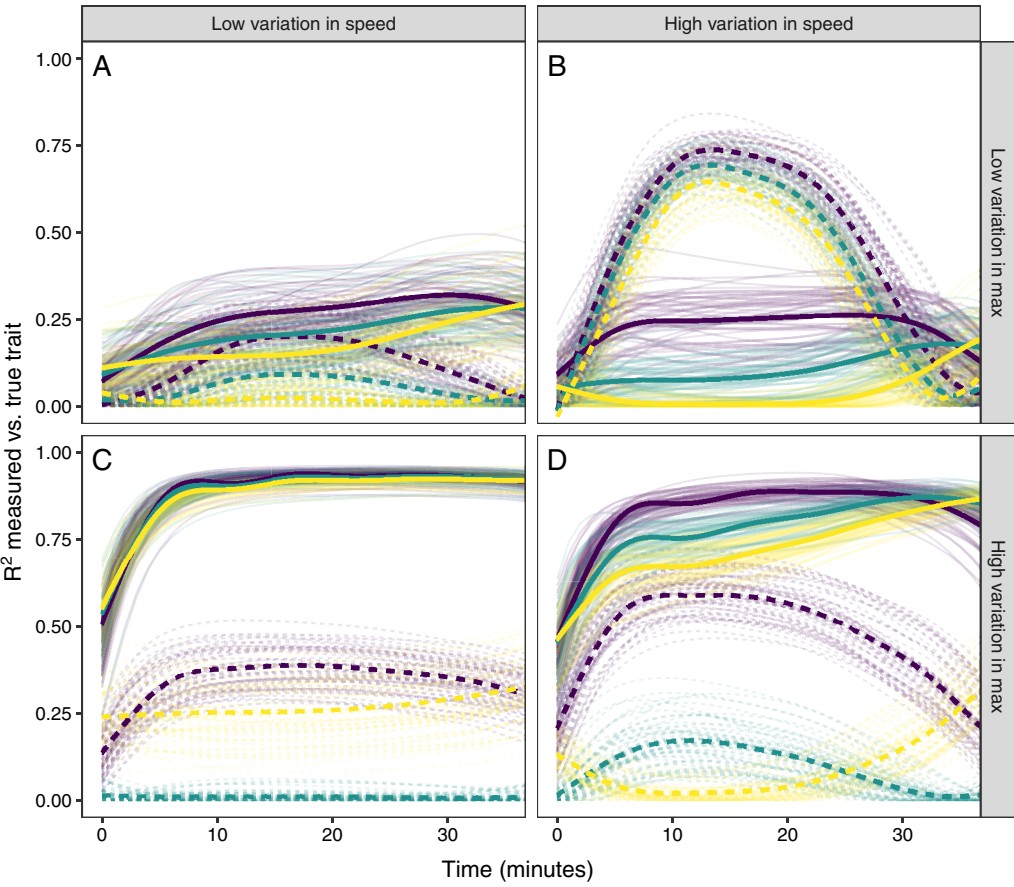

**Figure 6 Simulating different amounts of variation in speed and scope of response.** The relationship between single time point measures of glucocorticoids and the true value of either maximum level (solid lines) or the speed of the glucocorticoid response (dashed lines). Panels show results when the overall variation in maximum values and speed are both low (A), when one is low while the other is high (B and C), and when both are high (D). In each panel, three different simulation scenarios illustrate the patterns when speed and maximum value are positively correlated (purple), uncorrelated (teal), or negatively correlated (yellow). Faded lines show the results from each of 50 separate simulation runs and thick lines are the averages across all runs.                

glucocorticoid value and fitness does not differ for populations simulated with high or low between-individual variation (Figs. 7A and 7B). In all cases, however, the observed correlation is lower than the true correlation and always lowest in the population with low between-individual variation. The ubiquity of this pattern is a product of the simulation structure, because adding within-individual variation effectively adds noise to the true correlation. It is important to note that in the real world, it is unlikely that this pattern would be so universal, because unmeasured variables could influence both fitness and glucocorticoids. For example, if habitat quality directly alters fitness and glucocorticoids, the observed correlation could be stronger than the 'true' correlation. Thus, interpretation of these results should be made cautiously in light of the simplicity of the simulation compared to real world conditions. Nevertheless, general patterns illustrated by the simulation are likely to pertain across a wide range of conditions.
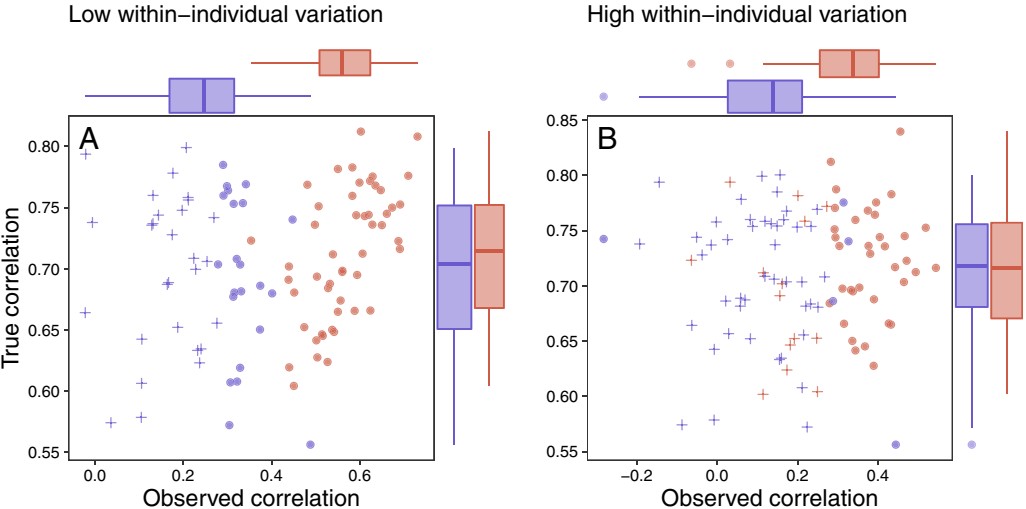

**Figure 7 Variation in response and ability to detect fitness correlations.** The relationship between observed maximum glucocorticoid values and fitness for simulated populations that have low between-individual variation in maximum glucocorticoids (blue) or high between-individual variation in maximum glucocorticoids (red). Each point is the result of a separate simulation of 50 individuals using the settings described in the text. Filled circles are simulations in which observed glucocorticoid values at 30 min were significantly correlated with fitness and crosses are simulations in which the relationship was not significant. Panels illustrate conditions with low within-individual variation (A) *vs* high within-individual variation (B). For each simulation, the correlation between true maximum glucocorticoids fitness is plotted on the y-axis and the correlation with observed values is plotted on the x-axis.

**Figure 8 Simulated data for a hypothetical control (blue) and treatment (orange) group.** Faded thin lines show the acute response for each individual simulated (20 per group) and thick lines show the average response curve for each group.

It was easiest to detect statistical evidence for known glucocorticoid-fitness relationships when within-individual variation was low (Fig. 7A). Under these conditions, the true pattern was recovered in nearly all simulated populations with high between-individual

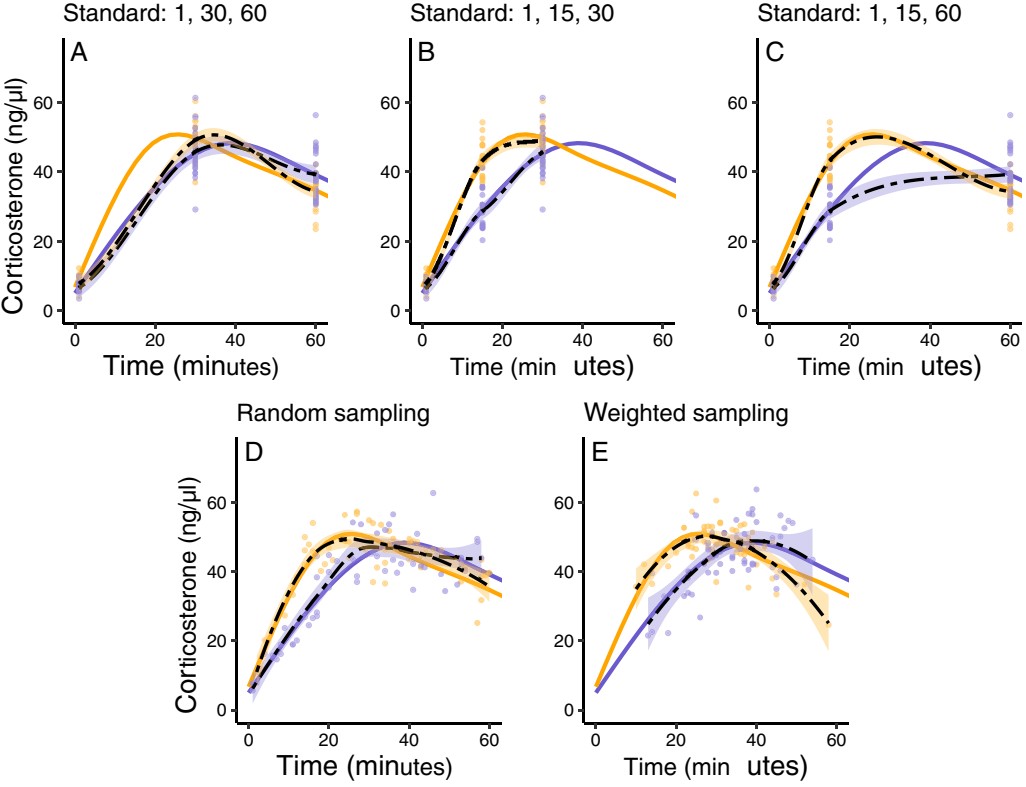

**Figure 9 Five possible sampling schemes to compare two groups.** For standard sampling (A–C), every individual is sampled at exactly three time points that include baseline along with samples at 15, 30, or 60 min. For random sampling (D) each individual is sampled at three random points between 1 and 60 min. For weighted sampling (E) three sampling times are chosen for each individual from a normal distribution with mean of 32 and sd of 9 min. In all panels, solid lines are the true group averages, dashed black lines are the estimates based on samples, shaded intervals are the 95% estimates from a simple generalized additive model, and points are individual samples collected.

variation and in approximately half of the populations with low between-individual variation. It becomes harder to detect these true relationships when within-individual variation is high (Fig. 7B) are high, but even in these more challenging situations the relationship can be detected the majority of the time if between-individual variation in maximum levels is high. When within-individual variation is high and between-individual variation is low, it is nearly impossible to detect glucocorticoid-fitness relationships.

## Scenario 5: designing optimal sampling strategies

In this simulated scenario, standard sampling schemes cannot fully describe the difference between treatment groups (Fig. 8; Figs. 9A–9C). Sampling at 1, 30, and 60 min completely fails to distinguish the difference between groups, despite the fact that the treatment group reaches it's maximum value on average 12 min (~40%) faster than the control group. Many empirical studies include an earlier sample at 15 min to capture the rate of initial increase (*Silverin, 1998*; *Cockrem, 2005*; *Cockrem, 2007*; *Cockrem, Potter & Candy, 2006*; *Huber et al., 2021*). Shifting one of the sampling time points in this

simulation to 15 min does recover a clear treatment difference in initial increase, but cannot capture the large difference in the time required to reach maximum values between the groups 9B-C).

In contrast, both the random sampling and weighted sampling schemes detect differences in the overall shape of the acute response (Figs. 9D and 9E). In this particular scenario, there is no clear difference between these two approaches and both perform well in describing both the difference in initial increase and the difference in the time required to reach maximal levels in each group.

## DISCUSSION

I demonstrate that simple simulation tools can produce datasets that closely match empirically observed acute glucocorticoid responses. Once developed, simulations can then be used to explore a wide range of hypothetical scenarios and to guide empirical studies. I explore a few possible scenarios with a limited range of parameters here. Nevertheless, even the simple demonstrations included in this article suggest several ways that simulation could help move studies of physiological flexibility forward. One of the main benefits of simulating datasets is identifying unmeasured properties and assumptions of currently available data that can become targets for future empirical work.

For example, I demonstrate that the ability to accurately measure a single glucocorticoid trait of interest (maximum glucocorticoid level) is strongly impacted by the amount of between-individual variation in the population and more subtly influenced by between-individual variation in speed (the time required to reach maximum levels) and by the timing of sample collection relative to the average population peak timing. In one sense, this result is unsurprising because it is intuitive that large differences in maximum glucocorticoids should be easier to detect, but there are important consequences of this fact for interpreting studies that seek to link between-individual variation in the magnitude of the stress response with other traits. The magnitude of the acute stress response often varies substantially across life history stages (*Wingfield, Vleck & Moore, 1992*). Even if study designs are identical it will be easier to accurately measure individual glucocorticoid traits under some conditions with the same level of sampling. This simple scenario shows that the same sampling regime will perform better or worse depending on the combination of glucocorticoid regulation parameters in the population being studied for statistical, rather than biological, reasons.

The pattern of within-individual covariation between separate aspects of the acute glucocorticoid response (*e.g.*, speed and maximum values) can also have important effects on the interpretation of empirical results. I found that covariation between speed and magnitude of the acute stress response and the relative amount of variation in each of these traits have profound effects on the ability to accurately measure either single component. This result occurs because measures taken at any given time point reflect a mixture of speed and magnitude. Empirical work specifically designed to assess covariation and variance at different times could help to understand what conclusions we can reasonably draw from available data. This simulation also clearly demonstrates that measuring timing components of the acute glucocorticoid response (*e.g.*, the time required to reach

maximum) is much more challenging over a wide range of conditions than measuring maximal levels. Although a great deal of empirical work has focused on estimating differences in the initial rate of increase in glucocorticoids (*Cockrem, 2005*; *e.g.*, *Cockrem, 2007*; *Wingfield, Vleck & Moore, 1992*), much less is known about between-individual variation in the time required to reach maximal levels.

Beyond the specifics of this particular example, what these results demonstrate is that understanding what aspect of the glucocorticoid response is being measured by any particular study design depends on extensive knowledge of the overall shape and amount of variation in different aspects of the acute stress response. There is still a great need for observational sampling to characterize within- and between-individual characteristics of glucocorticoid responses, particularly when new species or contexts are being studied. An increased focus on descriptive data collection can subsequently contribute to more targeted and powerful hypothesis tests and better parameterized simulations.

I also showed that the amount of within- and between-individual variation in glucocorticoid traits directly impacts the ability to detect statistical support for known relationships between the trait of interest and fitness. The fact that low between-individual variation and high within-individual variation in maximum glucocorticoids make it harder to detect true glucocorticoid-fitness relationships across a wide range of conditions has important consequences for interpreting empirical results. Many studies have demonstrated different relationships (or lack thereof) between glucocorticoids and fitness at different life history stages (*Bonier et al., 2009*; *Vitousek et al., 2018*), but it is also well known that the absolute amount of between individual variation in glucocorticoid traits varies considerably at different stages (*Wingfield, Vleck & Moore, 1992*). My simulation demonstrates that the power to detect true relationships will differ drastically across these conditions even with identical study designs and samples sizes; thus, apparent differences in glucocorticoid-fitness relationships across seasons or stages can easily arise as statistical artefacts when between-individual variation in hormones also differs across contexts. Great care may be needed to conclusively differentiate true differences in these relationships from statistical artefacts. Of course, it may be common for physiological traits to have stronger direct impacts on fitness in some contexts than others (*Bonier et al., 2009*), but simulation guided studies can help to ensure that the statistical approach to detect differences is equally powerful in different contexts.

Finally, I show that many standard sampling schemes perform poorly in some situations and for certain questions. A few clear takeaways can be derived from these results. First, while strict standardization of the timing of samples has some clear advantages, it also comes with costs and likely makes it nearly impossible to detect certain types of variation between groups or individuals. It should be clear that no amount of additional sampling would allow that approach to detect the difference in time to reach maximum glucocorticoid levels developed in the scenario here. Second, while it may be very difficult to accurately estimate the full shape of the acute stress response for individuals, alternative sampling schemes with random or weighted sample timing make it possible to describe these shapes accurately for groups (*e.g.*, treatments, species, different contexts) even without extraordinarily large sample sizes. A similar argument about the power of
randomly timed sampling has been put forward in the function valued trait literature (*Gomulkiewicz et al., 2018*), but this type of sampling scheme is rarely used in evolutionary endocrinology research (but see, *Vitousek et al., 2022*). It is perhaps unsurprising that empirical articles that have emphasized the importance of different time courses (rather than only maximum) of the stress response often focus on between group comparisons or investigate variation in the exact sampling time between individuals (*e.g.*, *Baugh et al., 2013*; *Small et al., 2017*). These alternative sampling schemes also come with drawbacks (*e.g.*, the inability to directly compare individuals sampled at different times) and I don't suggest that they are universally better options than standardization; rather, different schemes will perform better or worse given the specific scenario and the exact hypothesis being tested.

The simulation of different sampling schemes in particular addresses a single specific scenario, but a similar scenario could be designed for any number of studies and any number of predictions about how the speed, scope, or other attributes of the glucocorticoid response are expected to change with a treatment or between different groups or species. Clearly, when estimating the timing of peak glucocorticoids, a simple baseline plus induced sampling scheme is sub optimal, but this scheme may perform well in other situations where the maximum value is the target and there is relatively little variation in response time. Creating simulations like this before studies are conducted has the potential to increase the efficient use of researchers time and funds, but also forces researchers to think explicitly about quantitative predictions ahead of time. These simulations could be included as part of a study pre-registration, grant application, or registered report to demonstrate exactly what data collection and analysis approaches are planned and to justify those decisions. Across a wide range of disciplines there has been an increasing push for pre-registration, reproducible research, and transparent research practices (*O'Dea et al., 2021*). Simulation provides an opportunity for evolutionary endocrinologists to embrace these best practices by improving the quality of study design, allowing for more quantitative hypotheses and predictions, and providing a clear justification for experimental choices.

While there has been increasing interest in understanding within- and between-individual variation in the acute glucocorticoid response in recent years (*Hau et al., 2016*; *Lema & Kitano, 2013*; *Taff & Vitousek, 2016*; *Wada & Sewall, 2014*), the methods and data available to tackle these questions have changed relatively little. Many sophisticated statistical tools are now available and clear arguments have been made about the need to apply these approaches to endocrine traits, but relatively few empirical studies have effectively used these tools (but see, *Fürtbauer et al., 2015*; *Houslay et al., 2022*). Arguably, the biggest roadblock at the moment is the limited availability of empirical data needed to test hypotheses. Although there is a rich history of empirical work focused on understanding the acute glucocorticoid response (*Cockrem & Silverin, 2002*; *Romero, 2002*; *Vitousek et al., 2019*; *e.g.*, *Wingfield, Vleck & Moore, 1992*), many current questions require a level of repeated sampling within- and between-individuals that is difficult to achieve. Simulation offers one way forward, by allowing for more efficiently designed studies, by allowing researchers to identify when the question of interest can

in principle be answered with a given study design, and by providing a tool to translate recent theoretical advances (*Grindstaff et al., 2022*; *e.g.*, *Luttbeg et al., 2021*; *Taborsky et al., 2021*) into tractable, hypothesis driven empirical studies. Ideally, conceptual articles, empirical work, theory, and simulation will proceed together to make progress in this field. The tools presented here only scratch the surface of the ways that data simulation can be applied to address pressing questions in evolutionary endocrinology.

## ACKNOWLEDGEMENTS

I would like to thank Michaela Hau and her lab for discussions about the ideas presented in this project. John Wingfield also provided an insightful overview of the history of study design in field-based studies of the acute stress responses and pointed me to key references. Finally, I thank Maren Vitousek and her lab members for feedback and discussion on early versions of this project.

### Funding

Conor Taff was supported by NSF-IOS 2128337. The funders had no role in study design, data collection and analysis, decision to publish, or preparation of the manuscript.

### Grant Disclosures

The following grant information was disclosed by the authors:
Conor Taff: NSF-IOS 2128337.

### Competing Interests

The authors declare that they have no competing interests.

### Author Contributions

- Conor Taff conceived and designed the experiments, performed the experiments, analyzed the data, prepared figures and/or tables, authored or reviewed drafts of the article, and approved the final draft.

### Data Availability

All data and code are available in at GitHub and permanently archived on Zenodo:
– https://github.com/cct663/speed_vs_scope, Conor Taff. (2022). cct663/speed_vs_scope: v1.1 (v1.1). Zenodo. https://doi.org/10.5281/zenodo.6784203.
– https://github.com/cct663/simcoRt, Conor Taff. (2022). cct663/simcoRt: v1.1 (v1.1). Zenodo. https://doi.org/10.5281/zenodo.6784207.

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
