# Peer review of "Simulating physiological flexibility in the acute glucocorticoid response to stressors reveals limitations of current empirical approaches"

_PeerJ, doi:10.7717/peerj.14039_

## Round 0.1 · original submission · Major Revisions

The reviewers felt that your manuscript can be made acceptable for publication with some revisions. The reviewer comments are straightforward and should be responded to and the manuscript revised accordingly. In crafting your responses and making your revisions, please try to be more objective and state clearly the specific goals of your research and the specific conclusions.

Reviewer 1 ·

Basic reporting

This manuscript uses data simulation to generate realistic kinetics of acute glucocorticoid stress response. The simulated dataset then can be sampled at different time points to investigate how the estimated parameters would match the ‘real’ values. I think this is a very useful and timely approach that allows the critical evaluation of the ability of a given sampling protocol to uncover unknown patterns in glucocorticoid variation. In that respect, the paper advances the field by applying an important, yet currently scarcely used resource. Therefore, I liked the concept and I see merit in the paper. On the other hand, I see problems with the introduction, implementation, and interpretation of the study and I detail these concerns below.

Experimental design

(no experiment - see comments below).

Validity of the findings

(see below)

Additional comments

1. Introduction. While I applaud the concept and I approve of the justification, I do not think that the paper would fairly acknowledge previous efforts in analyzing acute glucocorticoid stress responses. There have been many approaches including conceptual papers, mathematical models, empirical studies (applying different sampling regimes, using reaction norms, etc.)… - basically none of which is cited here. This begs the question whether the author is not aware of this literature or just decided to disregard it. I think credit should be given where it is due, but a better review of the state of the art would also help focus the paper by clearly showing what is known and what remains unknown.
2. Implementation. While the functions used in the data simulation are available in a GitHub repository, I think the main functions and how they work should be also explained in the paper. The GitHub repo may or may not be available in the future, it may be superseded by more recent, developed code, etc. So while it is fine to provide the code there, I think it should also be described along with the published paper (e.g. as an appendix).
3. I also think that the structure of the manuscript and clarity of the writing would need some rethinking and improvement. The Introduction is somewhat repetitive (e.g. compare lines 64-66 vs lines 146-149), leaves key concepts unexplained (e.g. from lines 86 onwards, ‘FVT’), while the Results contain elements that would belong to the Introduction (lines 224-246), some to the Methods (e.g. lines 248-258) and others to the Discussion (many parts, including 430-440, 502-512). The current structure blurs and blends the questions, the actual methods to answer them, the results, and their interpretation.
4. One limitation is that the true parameters are drawn from a normal distribution, whereas endocrine traits may not always be normally distributed.
5. Interpretation. Some of the results would need a more thorough discussion. For example, while it is stated that low between-individual variation in the maximum glucocorticoid value may strongly limit our ability to detect true values and that this result is intuitive (lines 300-303), I would argue that it is actually not trivial at all why this should be the case. As between-individual variation converges to zero (i.e. all individuals showing the same phenotype), the model R2 value also approaches 0, indicating not only an imperfect fit, but a totally inappropriate model. Maybe I’m missing something here, but I think this is anything but an intuitive result.
6. Figure 7 is very hard to understand. First, it is not clear what blue and red colors stand for. The figure legend says: “…populations that have low between-individual variation (blue) or high between-individual variation (red)”. It is not clear variation in which trait (fitness or GC?). Second, if the red and blue represent groups with different amounts of individual variation, then how come different panels show situations with low vs. high individual variation? This really needs a better explanation.
7. I feel that some of the interpretations are flawed and are trying to prove a point rather than to provide an objective evaluation. I think the simulation presented in Figure 8 provides a good example. The parameters are deliberately chosen to make the point that using this particular sampling regime, it is impossible to detect the difference between the two treatments. However, the discussion (lines 546-558) goes further and states that “most study designs employed to date cannot _in principle_ detect those differences”. This is incorrect. From Figure 8. it can be clearly seen that a simple shift of the stress-induced sampling from 30 to 15 minutes should yield a clear difference between the treatment groups. Many studies included or even advocated an earlier sample before 30 minutes, for the very reason to be able to detect the initial steep rise in corticosterone (e.g. Cockrem 2005, 2007, Cockrem et al. 2006, 2008, Ellenberg et al. 2007, Eeva et al. 2005, Huber et al. 2021, Love et al. 2003, Silverin 1998 – just to name a few.)

Reviewer 2 ·

Basic reporting

Minor adjustments are needed. Please see 4. Additional Comments.

Experimental design

The simulations comment on future proposed experimental design. Simulations are well justified and do not overgeneralize. Please see 4. Additional Comments.

Validity of the findings

Raw Data and Code: I was able to navigate to GitHub site and while not entirely straightforward, I was, for instance, able to find Koolhaus raw data for rats. PeerJ journal editors should be sure to double-check and ensure raw data via GitHub and the way Github is accessed and displayed meets journal integrity.

The use of simulations is well justified, but I offer areas to reduce redundancy and clarify scope. Please see 4. Additional Comments.

Additional comments

Complete Reviewer Notes

Manuscript Author: Taff, Conor
Manuscript Title: Simulating physiological flexibility in the acute glucocorticoid response to stressors reveals limitations of current empirical approaches
Journal: PeerJ

Overall Importance: Stress sampling the same animal multiple times to examine within-individual variation in stress response is hard. This research provides a simulation model to do this repeated sampling. This simulation has developed user-specified characteristics and can vary key parameters including variation within and between individuals, relationship to multiple traits in the HPA Axis function (speed and maximum), and fitness. This simulation provides key insights to developing appropriate experimental design, especially for under-studied species and/or traits of the HPA axis. While I think that many researchers do their best running tightly controlled measurement practices, those researchers exploring relationships of traits or fitness or exploring and under-studied species where variation is yet unknown may benefit from these simulations when designing future experiments with awareness of possible methodological limitations.

Lines 35-40: These last two sentences seem redundant here. I’m wondering if they’d be better in place of earlier opening abstract lines or if they are not needed at all.

Lines 77-84: “..but that is currently not possible for most studies of endocrine flexibility”. Explain why to justify that it is not possible (lines 77-80)? Is it because you actually justify yourself in “reason #2” in lines 80-84? “…the functional shape of the response itself may be the important trait…” If so, these should not be two reasons, rather a singular statement of limitation/s with a couple of reasons why those limitations may be. You may want to re-work these lines as I don’t think “reason 1” is well-founded or at least not well-described alone.

Line 99-101: “…faces the same empirical challenges for within-individual reaction norms above, SUCH AS….” I suggest you add a brief “such as…” list.

Lines 112-117: I am wondering if you should state small vertebrate animals here. Seems like in larger wild animals, you could absolutely take more blood samples more often. Wondering if this “pitch” of this paper should be towards many of the avian and herp folks who use small animals to do stress work. Or perhaps repeated blood samples on a rhino or tiger aren’t done for lack of blood volume…but logistically just very hard to take for other reasons (ex. Sample size). Anyhow, this might be an area to add some real-world scope to draw in your intended audience quickly.

Lines 133-144: This is an extremely important and exciting portion of your manuscript. However, I do think you need to list or explain in order to justify. For instance, when you say “but don’t’ address the ways that the particular logistical challenges of hormone measurement might necessitate different empirical approaches”. Please list or explain some of those ways that need to be addressed and why.

Lines 146-159: Well done. An intriguing, justified and well-laid out example of importance and scope.

Lines 169: “…This simulation is designed to create data sets…”. I am unsure what you mean by create DATA sets. This is a simulation. So, are these created data sets simulated data paradigms? And would this be a method for those gathering empirical data from actual organisms to compare their real-world data to the simulated data that generates particular curves? You refer to “artificial datasets that have similar characteristics to empirical data for different systems.” in line 237. I believe this is what you mean, yes?

Line 201-202: “The function also generates a simulated performance (e.g., fitness) measure, based on the underlying true values.” Example on these “underlying true values” may be helpful here.

Lines 248-258: This is an important demonstration and helpful to show utility to the reader.

Lines 318: As a researcher aware of these sampling techniques, I think many avian stress biologists do try to control sampling of individuals in the same life history stage (i.e. Breeding, non-breeding, migrating, molting, etc.) and at the proper time point (30 minutes). It might be worth showcasing (in addition to Wingfield 1992) how your simulation helps to justify these good examples of targeted sampling in published studies. There must be a least a couple/few you could reference.

Lines 325-332: Well done. A timely piece to be explored. Indeed, there are many current researchers exploring these linked traits of the HPA axis.

Lines 364: Neither, nor.

Lines 360-369. This paragraph and subject matter seem rather…”oh of course”. If there is little difference between speed and maximum cort, of course it would be hard to capture differences. And if variation in speed, which is a rate, unsurprisingly is hard to “catch” using one sample (one data point). So, while this paragraph was not necessarily overly illuminating to me…is there an example that it could be applied to that does demonstrate how this scenario “could” illuminate something?

Lines 377. “…negative impacted” should be read negatively impacted. Yes?

Lines 380-382: “Beyond the specifics of this particular example, what these results
380 demonstrate clearly is that understanding what aspect of the glucocorticoid response is being measured by any particular study design depends on extensive knowledge of the overall shape and amount of variation in different aspects of the acute stress response.” I believe this gets to what I was mentioning earlier regarding Lines 360-369. Perhaps stating that this is particularly important for under-studied species rather than those that have been well and routinely studied and characterized regarding nuanced changes in HPA axis function across life-history stages, seasonal impacts, individual variation, etc. I think pulling together and demonstrating the impacts of lines 360-369 and 380-382 could clarify.

Lines 440: And one may also argue that fitness measurements can’t be stated ubiquitously for a species at one time point. I believe what you are saying (showing…with your model) is that some life-history stages may have larger impacts (or offer larger “constraints”) in achieving fitness. Thus, researchers must refrain from over-generalizing any findings of fitness correlations at one time point. Yes?

Lines 496-497. Very interesting for this field of study!

Lines 506-507. “may be perform well”. Do you mean may perform well ?

Lines 508-512: “Creating simulations like this before studies are conducted has the potential to increase the efficient use of researcher’s time and funds, but also forces researchers to think explicitly about quantitative predictions ahead of time. These simulations could be included as part of a study pre-registration, grant application, or registered report to demonstrate exactly what data collection and analysis approaches are planned and to justify those decisions.” However, it should be noted that in many under-studied species…the between or within individual variation will not yet be known. So, I researchers will need to start cautiously exploring getting a grasp on that variation before beginning. Thus, there’s still a lot of sampling (similar to Wingfield 1992) that needs to be done before applying this simulation. Or rather, I suppose your simulation will showcase what happens IF these researchers do not first determine the degree of possible within and/or between individual variation. Their data collected could be horribly skewed or not representative.

Lines 535-537: I believe here that you’re getting at what I’m describing above in my response to Lines 508-512. Good.

Lines 560-567: a bit redundant. I believe this has been eloquently covered in lines 523-528. I am wondering if you might consider rearranging and reducing redundancy.


Figure 1: I see what you’re getting at here, but would the figure look cleaner and more intuitive if all of the colored “peaks” were facing upright? The teal “maximum plateau” being upside down is a little odd-looking.

Figure 2: You refer to panels A, B and C, but they are not labeled as such in your figure. Please denote panels.

Figure 3: Fine.

Figure 4: Fine.

Figure 5: Fine.

Figure 6: Fine.

Figure 7: Fine.

Figure 8: Fine.

Figure 9: Very interesting, but this graph is hard to see solid vs. dashed lines on my end. Please rework.

Raw Data and Code: I was able to navigate to GitHub site and while not entirely straightforward, I was, for instance, able to find Koolhaus raw data for rats. PeerJ journal editors should be sure to double check and ensure raw data via GitHub and the way Github is accessed and displayed meets journal integrity.

---

## Round 0.2 · accepted · Accept

Thank you for your efforts in improving your manuscript in response to reviewer comments.

Reviewer 1 ·

Basic reporting

The author made a commendable effort to revise the manuscript, and I feel that the manuscript has been improved a lot in terms of clarity and coherence. In general, I think the author satisfactorily addressed my comments. I do not necessarily agree with every argument presented in the very detailed rebuttal letter, but that does not undermine the value of the work and should not be used to hold it back; these remain rather subjective and are more suited to a discussion during a coffee break at a conference. I think the present manuscript provides a very useful contribution and an analytical tool to the field, and I hope it will be used and cited widely by the community.

Experimental design

no comment

Validity of the findings

no comment